# Recent Progress in Testing and Characterization of Hardenability of Aluminum Alloys: A Review

**DOI:** 10.3390/ma16134736

**Published:** 2023-06-30

**Authors:** Chang Gao, Xin Liu, Dong Zhao, Yiming Guo, Shaohua Chen, Fantao Gao, Tianle Liu, Zhenyang Cai, Danyang Liu, Jinfeng Li

**Affiliations:** 1School of Materials Science and Engineering, Central South University, Changsha 410083, China213101021@csu.edu.cn (T.L.); lijinfeng@csu.edu.cn (J.L.); 2College of Mechanical and Electrical Engineering, Central South University, Changsha 410083, China

**Keywords:** aluminum alloy, hardenability, test method, characterization approaches, influence factors

## Abstract

In this paper, the progress of the test methods and characterization approaches of aluminum alloys hardenability was reviewed in detail. The test method mainly included the traditional end-quenching method and the modified method. While the characterization approaches of alloy hardenability consist mainly of ageing hardness curves, solid solution conductivity curves, ageing tensile curves, time temperature transformation (TTT) curves, time temperature properties (TTP) curves, continuous cooling transformation (CCT) curves, and advanced theoretical derivation method have appeared in recent years. The hardenability testing equipment for different tested samples with different material natures, engineering applications properties, and measurement sizes was introduced. Meanwhile, the improvement programmed proposed for shortcomings in the traditional hardenability testing process and the current deficiencies during the overall hardenability testing process were also presented. In addition, the influence factors from the view of composition design applied to the hardenability behaviors of Aluminum alloys were summarized. Among them, the combined addition of micro-alloying elements is considered to be a better method for improving the hardenability of high-strength aluminum alloys.

## 1. Introduction

Hardenability is traditionally used to describe the properties of a material as determined by the depth and hardness distribution of a sample’s hardened layer under specific circumstances [1,2]. The hardenability of a material generally depends on the quenching cooling rate employed on the pre-heated samples. When materials are subjected to quenching thermal field velocity changes, the hardenability is typically determined by the quench-hardened depth and the quench sensitivity. Many heat-treatable alloy materials are heated above their solvus line to form the supersaturated solid solution, which is followed by the formation of a nanometer-sized strengthening precipitate [3]. After that, a quick cooling process keeps the unsteady supersaturated solid solution present at relatively low temperatures. The cooling rate of a high-temperature saturated solid solution has a direct effect on the solute distribution state of an unsteady supersaturated solid solution alloy at a lower ambient temperature. It will be related to the evolution of solid phase transformation in unstable supersaturated solid solution alloys at a specific temperature, and it will also have a direct impact on physico-chemical properties, especially hardenability. The primary variables impacting the alloy’s hardenability, in addition to the solute elements added to the alloy, are the grain size, homogeneity, and original structure. The study on the hardenability of alloy materials is based on factors such as the testing process and characterization improvement, influencing factors, and more. It focuses on approaches to improve the hardenability of the studied alloys and clarifies the mechanism of the alloy’s high hardenability.

The hardenability of alloys has been thoroughly explored due to the alloy materials, specifically polycrystalline metallic materials combined with single or combined element addition, being widely used to suit the rising demand for engineering structural components. High hardenability materials have homogeneous mechanical properties in general and greater toughness and mechanical strength. When these materials are subjected to extreme strain or other conditions, the possibilities of deformation, cracking, and other mechanical phenomena are reduced. In general, the hardenability of non-ferrous metal alloys is better than that of ferrous metal alloys, since the phase transition of unsteady supersaturated solid solution non-ferrous metal alloys is not easy to produce in a solid state. For example, Al-Zn-Mg-Cu alloys are widely used in aircraft since they offer high specific strength, high specific stiffness, high toughness, as well as ideal corrosion resistance and machinability [2,3]. The Al-Li alloy is predicted to be widely employed in space-sky preparation in the future due to its excellent fatigue resistance, high strength, and superior mechanical homogeneity [4]. The bearing structural components consisting of high-performance Al alloys have been progressing toward large size and integrated preparation during the past few years in the aerospace industry. Additionally, stricter performance specifications for forgings with large cross sections and thick plates made of high-performance Al alloys have been proposed. The imbalance of hardenability brought on by hardenability is more pronounced as the alloy material sizes increase. The hardenability of an Al alloy is induced by the precipitation of the equilibrium phase on the scattered particle, grain boundary (GB), and sub-grain boundary (SGB) when the core of the thick plate is slowly chilled, which reduces alloy supersaturation and weakens the ageing strengthening effect [5]. For instance, as the thickness of aeronautical an Al alloy sheets rise, the problems of the hardenability impenetrability of the thick sheets’ cores become more severe since the low core cooling rate of sheets reduces their performance [6,7]. At present, high-performance Al alloys with higher hardenability are urgently needed to improve the service performance of the workpiece. Even though many researchers focus their attention on the study of the hardenability of Al alloy, a thorough and scientific system of evaluation for this property has yet to be developed.

The hardenability research of alloy materials originates from the industrial steel field, and the hardenability curves of industrial steel have mostly been determined. In this paper, the hardenability of Al alloys mostly refers to the hardenability test method of industrial steel including test devices, process flow, and evaluation methods. After all, current research methods for the hardenability of Al alloys are the same as or similar to those for steel [8]. Generally, the hardenability is measured by the end quenching-hardness test method, and the data are processed and characterized after the experiment. The distribution characteristics of the hardened depth layer and the evolution of hardness distribution of the test specimen can be used to express the hardenability characteristics of the tested material under the basic test conditions stipulated according to national standards or other scientific norms [3,8,9,10,11]. However, nonferrous alloy metals, especially high-performance precipitation-strengthening Al alloys, have not obtained the actual and complete hardenability curve in the actual process of hardenability testing and characterization [12,13,14]. Part of the results of the curve do not relate to the actual macro phenomenon of hardenability property [12]. 

At present, large-sized high-performance Al alloy structural components urgently applied in the aerospace field need more accurate hardenability properties data to quantitatively and accurately control the comprehensive performance of subsequent service performance. For example, the quenching characteristics of Al-Li alloys for large-scale structural components currently required for deep space exploration projects have a certain influence on the comprehensive properties of the alloy such as strength, hardness and residual stress distribution after treatment [15,16,17]. For these influence factors affected quenching behaviors, some works on improvement of the test and characterization of the hardenability of Aluminum alloys need to be developed. The special hardenability testing equipment was designed to suit the different types of alloys and quenching processes; the new theoretical derivation methods were developed to characterize the overall quenching change feature, and the hardenability of the determination method was systematically improved from the composition design of the quenching-resistance alloy by alloying and microalloying element control (shown in Figure 1). Hence, it is critical and meaningful to study the determination and characterization of the hardenability of high-performance Al alloy plates, as well as the influences of added element type and content on the hardenability property of Al alloys.

## 2. The Determination of Aluminum Alloy Hardenability

### 2.1. Traditional End-Quenching Method

Since the first few decades of the last century, the engineering field under the wave of the industrial revolution has continuously improved the requirements for the service performance of large-size metal structural components, which makes the exploration of scientific test methods for the hardenability of the metal materials plate an inevitable trend. The end quenching method was initially applied to the study of the hardenability property of steel, namely the top quenching test or the Jominy end quenching test [21]. Jominy and Boege hold initially applied the hardenability study of carburized steel in the 1930s. Due to the experimental object and operation process having extensive popularization easy handling and repeatability, it is still widely used in the process of all kinds of metal plate material hardenability research until now. These test procedures have been adopted by the world’s leading testing institutions and have formed a series of Jominy end quenching test standards, such as 1S0642, ASTM A255, SAEJ406, GB225, etc. [9,10,22,23]. The test samples in the Jominy experiment are mostly regular cylinders with a simple size design and convenient processing. In recent years, computers have been gradually used to simulate the changes in temperature field, tissue field, stress field, and hardness during the end quenching cooling test process [24,25,26], which makes the test operation more convenient and quicker. In the process of the end quenching method, the test samples of regular size are quickly clipped out and placed on the end quenching machine for temperature rising and heat preservation, and then the samples are taken out and cooling quenched in water flow on the device [12]. When the end of the sample is quenched, the cooling rate gradually decreases along the quenched end, and the corresponding microstructure and properties of the test sample change accordingly. A typical device consists of the frame, the high and low water tank, and the automatic circulating water system of the pump for measuring hardenability by end quenching. The hardenability characteristic curve of the correlation between the distance from the end quenching top and the changes in various properties can be obtained after the end quenching treatment of samples through the variety performance test [27], allowing for the evaluation of the hardenability of the tested alloy.

The device should theoretically be able to measure the depth of the quenched layer as well as the hardenability of the Al alloy during quenching cooling conditions. After the Jominy experiment was applied to Al alloy, there were some problems such as the incongruity with the actual application, the accuracy of the measured hardenability data, and the inability to obtain the fully hardened data. Numerous technical parameters assessment systems, including nozzle diameter, spray height, and sample size, which are suitable for an Al alloy hardenability test device, are not systematically and completely established to the field of hardenability evaluation of high-performance Al alloys [13,14,28,29]. Before quenching, the technological process and quenching fixtures differ from actual production, making it impossible for typical industrial quenching to accurately represent the quenching properties by the data obtained from the quenching cooling process [30]. The end-quenched sample was air quenched at one end while water-quenched at the other, which caused a discrepancy in the sample’s hardenability statistics. Since improper data conversion and characterization techniques were used to process experimental data after quenching, the Jominy test itself was unable to obtain fully quenched data. There are insufficient appropriate data conversion and characterization procedures used after quenching when processing the experimental data obtained from the quenching test of big-sized structural alloy. The data conversion was usually used to the relation between the quenching sensitivity and quenching hardenability by the mechanical properties change as a function of the distance from the quenching end, while characterization procedures were employed to determine the microstructural features evolution of the alloy materials sustained different quenching cooling conditions. As a result, the Jominy test itself was unable to acquire fully hardened data.

Some scholars proposed that the primality point was not meant to solve the hardenability measuring but refine the end quenching standard to suit the new type of alloy materials [31,32,33]. For example, before quenching steel alloy, it is necessary to carry out a conditioning treatment. During heat treatment, normalizing is needed to refine grains and homogenize tissues. The average temperature shall be specified according to the product standard, and the time of normalizing insulation shall be determined. In addition, the temperature is related to the hardening performance during quenching and affects the hardenability value, which should be specified clearly. The longer the heating and holding time is, the more uniform the austenitization composition of the fine-grain alloy structural steel is. The appropriate increase of grain can improve hardenability, so the standard of heating and holding time in the heating process needs to be unified. The above-mentioned problems and features of the end cooling method need to be considered as follows (as listed in Table 1). On the one hand, when no sample is placed on the support, the height of the water column above the nozzle mouth should be fixed to obtain a stable cooling rate on the test samples. On the other hand, the diameter of the water umbrella should be quantified to ensure the cooling area for the accuracy of the Jominy experimental data [24]. Furthermore, the mechanical tolerance defects of the sample during subtractive machining processing and assembly also affect the cooling performance of end quenching. For example, the verticality between the support surface and the vertical axis has a great influence on whether the quenching is biased so it should be constrained. The non-parallelism of the grinding surface affects the quenching effect as well as the positioning of the end quenching distance point, which requires accurate positioning [34].

### 2.2. New Modified and Developed Application of End Quenching Method in the Aluminum Alloy Field

At present, a wide demand for new high-performance Al alloy materials in industrial production gradually appears, which requires accurate hardenability data by modifying or improvement on traditional end quenching that the traditional method can’t obtain. In the hardenability determination of high-strength Al alloy materials, a modified Jominy test device is established by referring to the hardenability device for steel. Compared with other materials, high-strength Al alloy materials have greater thermal conductivity, so the circumferential heat transfer of end-quenched sample bars will affect the end-quenching experimental results, which should be eliminated. The elimination method is to add insulation facilities around the sample rod, or adopt the alloy round rod end samples of specific size for free water quenching, and control the end quenching time. The test device is shown in Figure 2 [15]. These measures not only eliminate the error caused by the high thermal conductivity of Al alloy material but also eliminate the adverse influence of circumferential heat transfer on the final test results.

In addition to the influence of strength properties on the Jominy experiment, different types of materials also affects the experimental hardenability data obtained by the Jominy end quenching method. High hardenability, medium hardenability, and low hardenability of hardenability differences of the tested materials will lead to differences in the internal organization after similar heat treatment. Therefore, the engineering performance of the material with different distribution characteristics of the strengthened precipitations is inconsistent. These three factors including hardenability, mechanical properties and microstructure properties are related to each other. A method of top quenching is designed to measure the relationship among these three related factors, which is also suitable for Al alloys. Generally, top quenching equipment is used to quench round steels or other alloys as samples. In the modified top quenching, the height of the samples is changed to half of the thickness to become flat square samples [35]. The observation of mechanical properties and fracture conditions can be facilitated in this quenching way. For medium and high hardenability materials, since the strength that can be used is still very high, the same strength and tempering temperature are selected to compare the hardenability of the material. The performance difference at different distances from the top after tempering and draw the strength curve is first measured. The top quenching method relates the hardenability of the material, the mechanical properties and the condition of the top quenching structure. For Al alloys with high hardenability, it is difficult to compare the hardenability differences through the traditional Jominy test. So, a new method to attach importance to thermal insulation is designed to accurately compare the differences between hardenability. In the modified end quenching method, a stainless-steel thermal insulation pressing block was added to the air-cooled end face of the Jominy sample and making the water jet to the lower tip of the sample after heating [36]. During the modified testing process, the cooling rate of the sample changes continuously measured by the hardness curve drawing with the half-cooling time to compare the hardenability difference of different materials (shown in Figure 2).

The accurate and efficient determination method of hardenability is not only needed to determine the strength and hardenability of Al alloys in the laboratory environment but also needed to accurately reflect the hardenability in the monitoring of the actual production process. To prevent the negative influence of non-uniform distributions of the particles in the actual production of the high-performance Al alloy, the hardenability of non-standard size samples is considered to measure to show the actual hardenability of the studied alloys. For measuring the hardenability of non-standard Al alloy, it is necessary to ensure the one-dimensional heat transfer state. Designing the non-standard end quenching test device, as well as carrying out contrast experiments on standard and non-standard samples to verify the correctness of the measurement device. In the comparison test, the holding time of standard samples is longer than that of non-standard samples at a certain temperature. The standard samples are quickly produced and moved to the end quenching testing machine for post-quenching cooling. The non-standard parts are equipped with thermal insulation sleeves, which is similar to the device applied to the high-strength Al alloy. The thermal insulation sleeves are designed as a steel pipe, the diameter of which is larger than the standard sample. The new insulation unit, aluminum silicate fibers loading between the steel pipe and the sample, is shown in Figure 2, which is designed with accounting for the factor of small mass volume of non-standard parts [16]. In the new device, reducing the insulation time on the non-standard sample achieved the same insulation effect as the standard samples.

In addition, the side heat dissipation rate of non-standard samples is much larger than that of standard samples. To ensure one-dimensional heat dissipation and reduce side heat dissipation of non-standard samples, it was equipped with internal and external thermal insulation sleeves to ensure homogeneous thermal conductivity. When the one-dimensional cooling rate of the non-standard samples is similar to that of the standard specimen, the obtained end quenching curve is similar. The test device can be used to determine the hardenability of the end quench specimens, which are not made to a standard size. Table 2 shows the new end-quenching methods used to compare different properties of Al alloys and different Al alloy samples. It could be seen that treating the sample with adequate heat preservation, controlling the reasonable shape and size of the sample, selecting the correct quenching temperature and quenching and holding time generally has a positive influence on the accurate determination of hardenability measurement.

The Jominy end quenching experiment applied to the aluminum alloy field has more problems, such as inaccurate data, complicated steps, and the fact that it is time-consuming. To solve these problems, researchers from various countries have not only developed special testing methods for specific properties of Al alloys but also improved the process flow and device of the end quenching method. In the traditional end-quenching process, there are two obvious problems. One is that the data of the two test surfaces of the sample are different. The other is the hardness distribution gradient at the small to the large distance between the same test surface and the end face doesn’t properly conform to the theory of the large to small distribution. The above-mentioned problems were magnified by hardenability measurement applied in the ending quenching test of the Aluminum alloy. Therefore, the end quenching process is need to improve in many aspects as follows. (1) The key to the success of the end quenching test is that the test plane should be taken at the same distance from the original sample surface. During the quenching process, the nozzle should spray water at the centre of the end of the middle sample to prevent deviation of the data of the same distance between the two test surfaces. (2) The flange is designed at the non-quenched end of the sample to prevent the oxidation of the pattern from contacting with the air in a large area. (3) When the testing sample was heated to a certain temperature, the steel can was necessary and designed with the appropriate amount of carbon powder at the bottom to completely prevent the decarburization of the pattern. (4) During the hardness test, ensure that the test point is on the centre line of the sample and keep the distance stipulated by the standard, to prevent the data difference on the test surface [11]. The test process is shown in Figure 3. The improved end quenching process prevents oxidation and decarburization of samples, ensures the accuracy of hardness values, and the hardness distribution gradient conforms to the theory.

To overcome the problems of the traditional end quenching device on complexly using, low mechanization degree and low efficiency, some kinds of new experimental methods are designed, such as placing the workpiece on the support plate, heating the workpiece tested in the same quenching device, and cooling the workpiece on the hose water outlet [37]. In addition, the data acquisition device is designed to quickly obtain the hardness value of each test point [38]. These new methods have the obvious advantage is that less manual intervention, higher mechanization degree and experimental efficiency.

To solve the current problem that only one sample can be quenched at the same time by the traditional end quenching device, some new quenching devices were designed for the hardenability testing of Al alloys needed two or more samples for the end quenching experiment. Meanwhile, the total number of single only one sample end quenching experiments is unreasonable due to the different quenching transfer times and water spraying amount by using the traditional end quenching device. While manual errors will be introduced into the cutting of samples after quenching treatment. Sun [39] et al. invented a new end quenching device, which fasten two types of Al alloys on a fixed frame, installed a U-shaped tube at the bottom of the pattern, and set a water inlet at the bottom of the U-shaped tube. The device can simultaneously measure the hardenability of two Al alloy samples and obtain a wide range of cooling rates. The experimental variables due to the two types of tested alloys were controlled, the experimental efficiency was improved, and the error caused by the different experimental conditions of aluminum and steel was eliminated. Li et al. [40] invented an instrument using a combination of metal sheet and end quenching for determining the hardenability of Al-Li alloy by laminated end quenching of many thin plates. The mechanical tensile property of the sheet combination of the testing alloy sample after the quenching method is tested to determine the hardenability. The method could simultaneously determine the hardenability of multiple Al alloys with different components. It does not need to be cut in the subsequent processing to avoid the introduced machining error. In this new end-quenching method, the hardenability test of Al alloy materials was creatively linked with the more precise tensile testing compared to the hardness testing, which made the hardenability evaluation system of Al-Li alloy more diversified, as listed in Table 3.

In the actual production process, it is also necessary to determine whether the quenching temperature meets the requirements of profiles and wall thickness at all times, to avoid material scrap at a higher applied temperature. Hence a device is designed to accurate feedback and adjustment of quenching temperature during the hardenability test of metal materials. In the device, the water spraying unit and drainage unit, in the end, quenching device are installed on the test cylinder [41]. In this way, the hardenability of Al alloys and other materials was facilitated and safely analyze at the control high quenching temperature, which could provide a basis for the rational design of a quenching temperature system in a wide temperature range. Meanwhile, these innovative testing methods improve the process flow and testing methods from various angles of the process design. The new instruments and processes are suitable for testing the hardenability of various Al alloy materials, as shown in Table 4. These devices and methods improve the data accuracy, test efficiency, and mechanization degree of the end-quenching comprehensive performance, making new progress to the traditional end-quenching method.

In addition to the traditional Jominy test and the modified test, some non-customary methods, such as the critical diameter method and the borehole plug method were applied to determine the hardenability of the tested alloys. These methods do not have universal applicability and can only be used to determine the hardenability of specific types of Al alloy materials. When the predicted hardenability of the sample is relatively large, the “drilling and plugging sample method” can be used, which can accurately reflect the core of the intact sample of the same size through the changes of the small sample in the centre of the hole during heating and cooling. The technique may address the issue of erroneous hardenability values when identifying materials with high hardenability [42]. Although the main hardenability test methods are still based on the Jominy end hardenability method, the application scope of these improved new methods has been expanded to meet the needs of more hardenability test scenarios.

## 3. Methods for the Hardenability Characterization of Aluminum Alloys

The data analysis obtained from various hardenability test methods can be used for characterizing the hardenability of Al alloys, helping to select the appropriate work piece manufacturing process for different alloys of different processing sizes for different hardenability requirements. At present, in addition to characterizing the hardenability of Al alloys based on the traditional mechanical properties testing related to the quenched depth, many new hardenability’s characterizing methods and models appeared. The characterization of the hardenability behavior of Al alloys can be evaluated utilizing hardness curves, tensile strength curves, solid solution conductivity curves, TTT curves, TTP curves, CCT curves and other special theoretical calculations, as shown in Table 5. These characterization methods provide the basis for the accurate evaluation of the hardenability of the Al alloy from different points of view of analysis.

### 3.1. Hardness Curve and Tensile Strength Curve

The traditional end-quenching method applied to alloy materials is generally characterized by hardness value. Hardness value changes as a function of the distance from the end is tested and summarized to characterize the hardenability of metal materials. The traditional method is the most common and easy to visualize for standard samples. After quenching and ageing heat treatment, the hardness values at different positions away from the quenching end are measured. After the ageing hardness data are obtained, there are usually two methods for analysis and processing. One is to select the value points with the same hardness variation trend compared to the distance from the quenching end. This method was used to map the ageing hardness curves of Al-Zn-Mg-Cu and other quenched Al alloys at room temperature [15].

To compare the strength and hardenability of Al alloys, the depth of alloy quenching was defined as the distance at 90% of the maximum hardness of the quenched alloy. The data analysis to strengthen hardenability is more accurate due to the small interval of sampling points and the data fitting with rational functions. Another method is to intercept same-length samples at the same distance from the quenched end and compare the hardenability changes of Vickers hardness at both ends [13,43]. If the Vickers hardness variation of the sample is small, the hardenability is better. The method controls the variables of the hardness test and ensures the accuracy of the experimental results for hardenability measurement. Hence, it could visually compare the hardness change of various metals, to judge the strength and hardenability of the alloy with different quenching depths. The hardness curves obtained by these methods characterize the strength hardenability of Aluminum alloy and the mechanical properties related to hardenability properties.

With the development of science and technology in the material field, the demand for large aluminum alloy structural components in aerospace and other fields gradually increases. The traditional hardness determination cannot meet the hardenability accuracy characterization of large-size structural parts, so the tensile strength characterization method is introduced in recent years. The method is represented by the “laminate end quenching method” to plot tensile strength curves to characterize hardenability behaviors [19]. The hardenability can also be further characterized by secondary processing of the obtained tensile strength curve by calculating the strength decline rate [29]. Among them, the “lamination end quenching method” is usually applied to the hardenability characterization of high-strength Al-Li alloys. The data were obtained from a mechanical tensile test and then drawn with the tensile strength curve to determine the hardenability [19]. A total of 95% of the maximum value of the ageing hardness can be selected as the quenched hardness of the alloy. The corresponding distance from the quenching end is the quenched depth of the alloy in the method. This method requires several tensile tests to obtain the tensile strength curve, which takes a long time and consumes a lot of sheet metal in the material processing and testing. However, in the method, the difference between the central layer and the surface layer can be reflected by measuring the change in tensile strength of different thin plates at the same position. By measuring the tensile strength of the same plate at different positions, the relationship between the retention fraction of tensile strength and the distance from the quenched end can be quantitatively determined. For example, Liu et al. [44] conducted hardenability test on the 2060 Al-Li alloy by this method, and obtained tensile strength variation curves at different distances from the quenched end, thus evaluating the hardenability of the high-strength Al-Li alloy. Other than that, the mechanical hardenability behaviors (including hardness, tensile strength, and retention fraction) of some other high-strength Al alloys by the lamination end quenching method were summarized and are shown in Figure 4 and Figure 5.

### 3.2. Conductivity Curves

The conductivity value change as a function of the distance from the quenching end can objectively reflect the quenching sensitivity of the alloy. The quenching conductivity curve with the distance from the quenching section can be mapped with the quenching end of the Al alloy at room temperature to characterize the hardenability behaviors [11]. As shown in Figure 6, many corresponding kinds of research on the conductivity curve reflecting the hardenability property applied on the high-performance Al-Cu-Mg alloys and Al-Zn-Mg-Cu alloys are summarized and displayed. Generally, the solid solution conductivity of Al alloys reflects the saturation of solute atoms in the supersaturated solid solution. According to Mathieson’s resistivity law, the higher the degree of supersaturating of the solid solution, the greater the degree of distortion caused by the solute atoms dissolved in the lattice of the Al matrix [45]. The number and density of electron scattering sources in the matrix increase, the average free path of conducting electrons decreases, the conductivity of the alloy decreases, and the hardenability of the alloy decreases. The volume fraction of supersaturated solid solution cooled to room temperature in the tested alloys with low quenching sensitivity property is large, while the conductivity of supersaturated solid solution changes little with the measured variable. It reflected in the conductivity curve of the solid solution state, the relatively smooth line was linked to the small degree change of the hardenability. The method is analyzed from the point of view of the physical properties of the studied alloy. It is often used together with the age-state hardness curve to evaluate the hardenability property of the targeted Al alloy, which can be mutually verified to prevent a large deviation from the experimental results.

### 3.3. Time-Temperature-Transformation (TTT) Curve

The first isothermal transition curve of the overcooled austenite was determined by Daveport and Bain in the 1930s. Many researchers focused on the isothermal transition curves of various alloy materials. The isothermal transition curves mainly consist of some C-shaped curves, which were known as the C curve. It is currently called time-temperature-transformation (TTT) curve, also known as the TTT curve.

The TTT curve is a type of isothermal transition curve, which can be obtained by measuring the transformation conversion of alloys at different temperatures. For example, JMat-Pro 8.0 software is used to simulate the TTT curve of η phase transition of Al alloys with different Zn contents [29]. TTT curve can also be drawn based on quenching conductivity and hardness values of Al alloys [23]. TTT curve reflects the phase transition process of the material under different undercooling degrees. The quenching sensitivity of Al alloys can be evaluated comprehensively by observing the nose tip temperature, holding time, and other important data of the TTT curve.

### 3.4. Time-Temperature-Properties (TTP) Curve

The time-temperature-properties (TTP) curve is an isothermal age-performance transition curve, by which the isothermal time to achieve a specific performance fraction at a specified transition temperature can be identified [51]. TTP curve is one of the most common means to study the hardenability of Al alloy. The value data of the TTP curve can be obtained by the property testing of the quenching of the end of the alloy. For example, as shown in the TTP curves of Al-Zn-Mg-Cu alloy in Figure 7, stage quenching was employed on the hot-rolled Al alloy plate cut into a sample of a specific size to determine the hardenability behaviors [14]. In these diagrams, it was found that compared with single-stage quenching, the TTP curve of the alloy after two-stage quenching shifted to the right. As known to all, the closer the nose end of the curve is to the right, the lower the quenching sensitivity and the better the hardenability. Therefore, the Al alloy adopts two-stage ageing to reduce quenching sensitivity and improve hardenability [26]. In addition, it can be seen that the incubation period of the Al alloy is relatively short on the obtained “C” curve compared to the steel alloys. While the latent period corresponds to the beginning of rapid precipitation, and the alloy with a short latent period has high quenching sensitivity. In these TTP curves, the high-strength 7055 Al alloy with a higher alloyed element content has higher quenching sensitivity [46].

### 3.5. CCT (Continuous-Cooling-Transformation) Curve

The continuous-cooling-transformation curves generally reflected the transformation rule of supercooled austenite under continuous cooling conditions, which is close to the actual cooling condition of heat treatment in the material processing. Therefore, it becomes an important basis for analyzing the transformation process of austenite and the microstructure and properties of transformation products during the continuous cooling process. It is more difficult to the measurement of the CCT curve than the TTT curve due to the difficulty to maintain a constant cooling rate, to quantify the microstructure in the transformation mixture, and to measure the time or temperature during the rapid cooling process.

The methods of measuring the CCT curve include the DSC test, dynamic resistance test, XRD test, and hardness test. Among them, the dynamic resistance method usually draws the resistance-temperature curve, and draws the CCT diagram of the cooling transition according to some important data of the cooling curve, to mainly characterize the hardenability [47]. For accurately determining the hardenability of the aluminum alloy, Jmat-Pro 5.0 software can be used to simulate the mass fraction of the CCT curve combined with the main metastable phase produced during the heat treatment process [17,52]. For example, in the summarized diagrams of the CCT curve of the aluminum alloy, the larger the mass fraction of the particles in the tested alloy, the smaller the hardenability (as shown in Figure 8). The method also uses simulation and analysis software to fit the obtained experimental data. In the obtained fitting CCT curves, a wider temperature range and a more obvious trend of continuous cooling transition are shown to evaluate the hardenability evolution of the tested alloy materials.

### 3.6. Theoretical Calculation Method

With the development of science and technology, the hardenability data measured by experiments is closer and closer to the real property evolution in the actual processing. However, the calculation methods as a complementary means to accurately characterize the hardenability, are increasingly becoming the primary means of hardenability performance testing. The theoretical calculation method can use the cooling rate curve, TTP curve, cooling curve, etc., to calculate the hardenability through particular formulas. According to the report by Liu [48], when the aluminum alloy is quenched in different media after a solid solution, different cooling rates (generally 1~1000 °C/s) can be obtained, and the cooling rate curve can be obtained. The formula for calculating the performance degradation caused by the decrease in cooling rate is as follows:Q = (P[F] − P[S])/P[F] × 100%

Q is the degree of performance degradation caused by the decrease in cooling rate, P[F] is the performance when the cooling rate is the fastest (usually can be quenched by water at room temperature), P[S] is the performance at a lower cooling rate (such as 100 °C water, oil or air quenching). In the formula, the greater the Q value is, the greater the loss of the alloy performance is with the decrease of the quenching rate, which reflects the higher the quenching sensitivity of the aluminum alloy. Determining the hardenability of Al alloys by calculating the cooling rate formula is the simplest way to evaluate its quenching sensitivity at present. The method of using the TTP curve and cooling curve to predict the influence of continuous cooling on Al alloy corrosion and yield strength is called Quench factor analysis by Evancho and Staley [53]. The formula in the calculation analysis is used to describe the C-curve as follows:$$C_{t}(T)=-k_{1}k_{2}exp[\frac{k_{3}k_{4}^{2}}{{RT(k_{4}-T)}^{2}}]exp(\frac{k_{5}}{RT})$$

In the formula, Ct(T) is the critical time required for precipitation of a certain amount of solute, k_1_ is the natural logarithm of the untransformed fraction, k_2_ is a constant related to the reciprocal of the number of nucleation, k_3_ is a constant related to the nucleation activation energy, k_4_ is a constant related to the temperature of the solid-solution phase line. While k is a constant related to the diffusion activation energy. R is the gas constant (8.3143 J/(K·mol)). T is the thermodynamic temperature.

The quenching factor τ was also obtained by the following formula. In the formula, t0 is the start time of quenching; t_f_ is the end time of quenching; Ct(T) is the critical time corresponding to different temperatures of the C curve.
$$\tau =\int_{t_{0}}^{t_{f}}\frac{1}{C_{t}(T)}dt$$

According to the report by Liu [54], the method is used to predict the hardness curves of 7055 Al alloy linked to the hardenability behaviors after ageing treatment. It was found that the lower the cooling rate of the tested alloy, the greater the quenching factor value, and the smaller the calculated hardness value, which was used to reflect the hardenability of the material. In addition, the hardenability of Al alloys can also be characterized by mathematical models and calculations. A thermodynamic model of microstructure development and room temperature hardness of steel during heat treatment is established by Li [49], which can also be used to predict the properties of Al alloys. The thermodynamic model consists of the finite element model which causes heat transfer and the reaction kinetics model which causes austenite decomposition. The experimental method is applied to quenching the end of the Jominy sample. The model based on the Kirkaldy model [50] reconstructed the reaction kinetics model of austenite decomposition and expanded the range of applicable chemical components. After the thermodynamic model is applied to the Jominy hardness prediction of Al alloy, the prediction accuracy is obviously improved compared with the original Kirkaldy model. In general, this is a new method used to evaluate the hardenability of Al alloys by numerical calculation, which also provides a model for the construction of other mathematical models of metal alloy hardenability.

## 4. Effect of Composition Design on the Hardenability of Aluminum Alloys

The mechanical properties of Al alloys, especially high-performance alloys consisting of Al-Cu alloy and Al-Zn-Mg-Cu alloy, are effectively enhanced by some type and different content of element addition. The content of elements in the Al alloy was determined to affect the hardenability of the alloy. The influencing factors include the difference in the content and proportion of main allotment elements and micro allotment elements [55]. Therefore, precise control of the content of various elements in alloys has a profound impact on the application of Al alloys, such as the manufacture of large structural components in aerospace. For example, Al-Zn-Mg-Cu alloys with high strength and toughness mainly used in the aerospace field are composed of Al, Zn, Mg, Cu, and other elements. The content and proportion of these elements have an important effect on the quenching ability of Al alloys. The relatively higher content of Zn and Cu in the Al-Zn-Mg-Cu alloy was the main reason for different hardenability behaviors compared to the Al-Cu alloy with low Cu and Zn content. Therefore, it is significant to summarize and clarify the effect of these alloy elements on the hardenability of high-performance Al alloys for designing new high-hardenability Al alloy materials.

### 4.1. Effects of Main Alloying Elements on the Hardenability of Aluminum Alloys

The influence of main alloy elements added into Al alloys on hardenability consists of Zn, Mg, Cu, etc. Zn element is one of the main components of Al-Zn-Mg-Cu alloy, which is widely applied in the aerospace industry. The immersion-ended quenching method [56] is used to study the influence of Zn content on the quenching sensitivity of Al-Zn-Mg-Cu alloy with different Zn contents within a certain range, as shown in Table 6. Based on the research, the hardness reduction of air-quenched and water-quenched Al alloy samples was compared by our team [19,31]. In additional, Jiao reported that the differences in the quenching sensitivity of alloys with the changes in quenching rate and Zn content are also explored [56]. According to the results of the above-mentioned studies, the quenching sensitivity of the alloy can be improved with the decrease of Zn content when the contents of Mg and Cu are approximate. When the content of Zn is constant and the quenching rate is high, the quenching sensitivity increases first and then decreases. While the quenching rate is low in thick plate alloy materials, the quenching sensitivity of Al alloys with higher Zn content is greatly improved.

The Mg element is another major alloying element in Al alloys. With the increasing of Mg content to a certain extent, the alloy in the slow rate of quenching could get more strengthen phase [3,6,31], which was linked to the hardenability behaviors. Meanwhile, as the Mg addition, the T (Al_20_Cu_2_Mn_3_) phase and S (Al_2_CuMg) phase are precipitated during the heated treatment processing. The quenching sensitivity of Al alloys increases with the decrease of Mg content and decreases with the increase of Mg content. Xi et al. [57] found through the method of end-quenching combined with the hardness test, the maximum hardness value of Al alloys increases with the increase of Mg content, but the depth of ageing strengthening decreases. By studying the influence of the content of Zn element and Mg element on the hardenability of Al alloy, it is found that there is a critical point of element content, related to the best hardenability at the point. However, the evolution rules of linear change of hardenability with specific element content have not been studied, and its internal influence mechanism needs to be further studied.

Different scholars also analyzed the influence of Zn and Mg on the hardenability of Al alloys by various quenching experiments. When exploring the influence of the mass ratio of Zn and Mg on Al alloy, a series of experiments were carried out in combination with end-quenching, room temperature water quenching and air quenching, observation of microstructure and atlas, hardness test, tensile test, electrical conductivity test, and metallographic microscope of hardness tester [58,59,60,61,62,63]. It is found that when the quality of Zn and Mg is relatively high in the studied alloy, the quenching depth of the Al alloy bar is positively correlated with the content of Mg. At the same time, the sample with high content of Mg has low hardenability [59]. In addition, with the increase of the mass ratio of Zn and Mg, the depth of quenching increases in turn in the alloy quenching samples [61]. The quenching sensitivity decreasing of the alloy was related to the expansion of the alloy with a complete quenching ɑ area, which inhibits the quenching precipitate phase. While the alloy with a high Zn/Mg mass ratio has higher Vickers hardness and lower quenching sensitivity than the alloy with a low Zn/Mg mass ratio. In summary, the content ratio of Zn and Mg has more effect on the hardenability of an Al alloy than the single element addition. The higher the Zn/ Mg content ratio of the Al alloy, the lower the quenching sensitivity, and the better the hardenability.

The Cu element is another important constituent element of high-performance Al alloy. The effect of Cu content on the properties of Al alloys was studied by the fusion casting, homogenizing annealing and end quenching experiments [59]. As while the influence of Cu content is researched by metallographic microscope, transmission electron microscope, scanning electron microscope [61] and differential scanning calorimetry [60]. The hardness of Al alloys with different Cu contents was tested and the hardenability of the alloy was characterized by the end quenching method. It can be seen that as the content of Cu is increasing, the depth of the quenched layer of the Al alloy decreases [60]. The hardness retention value of the alloy decreases with the increase of the distance from the quenched end surface. Jiao [56] reported that the hardenability decreases with the increase of Cu mass fraction. The hardness and hardness retain the value of the high performance with different Cu content were displayed and shown in Figure 9. In addition to exploring the influence of single Cu content by the traditional end quenching method, the immersion end quenching method can also be used to explore the relationship between quenching rate and Cu content [58]. It was found that with the increasing quenching rate, the content of Cu almost does not affect the quenching sensitivity. while the quenching rate is low, the quenching sensitivity increases with the increase of Cu content [61].

The influence of the interaction between the three elements on the hardenability of the alloy is obtained by the relation of the main alloying element content of Zn, Mg and Cu related to each other during the quenching process. For example, when the mass ratio of Zn and Mg is constant, the increase of Cu content reduces the effect of ageing strengthening and hardenability [3,31]. In the state, the reduction of Cu content can reduce the distortion of the supersaturated solid solution, so the stability of the solid solution increases and which enhances the hardenability of the alloy. Moreover, by observing the microstructure characteristics of the material, it is found that when the quality of Zn and Mg is relatively high and the content of Cu is low. It is conducive to inhibiting the quenching precipitated phase and reducing the stability of metastable η′ phase in Al-Zn-Mg-Cu alloy, the hardenability consequently could be improved [60]. According to the characterization results, when the content of the Zn element remains unchanged, the hardenability improves with the increase of the mass ratio of Cu to Mg [57]. Therefore, Zn, Mg and Cu elements interact with each other, and the change in the mass ratio of any two of them will lead to a change in hardenability. In the composition design of the actual production process, the best quality ratio of these three elements according to different property requirement need to be determined by reasonable hardenability test method and characterization, to ensure the best hardenability performance of structural parts.

### 4.2. Effect of Trace Elements on Aluminum Alloy

Compared with the main added alloying elements, the proportion of trace added elements in the Al alloy is usually less than 1 wt.% in mass percentage. However, some of them still have a great influence on the quenching sensitivity of Al alloy. Among them, the elements that have a greater impact on hardenability include Cr, Zr, Mn, Ge, Sc, Fe, Si and so on. For example, in the observation of the microstructure of Al-Zn-Mg-Cu alloy containing trace Cr, Zr and Mn, it is found that the quenching sensitivity of Cr-containing Al alloys is related to the precipitation of a large number of coarse-η phase particles nucleated in Cr-rich phase [62]. Further study found that the degree of influence of Cr, Zr and Mn on quenching sensitivity was arranged in the order of Cr, Mn and Zr from large to small. In addition, the influence of Ge and Sc content on hardenability was also found that with the increase of Ge content, the grain size of Al alloys is refined and the high melting point phase is formed, which significantly reduces the quenching sensitivity. Moreover, the vacancy density of the studied alloy during the quenching process is reduced, which inhibit the formation and growth of heterogeneous nucleation. So the hardenability of Al alloys is improved by the increase of the precipitation hardening phase after ageing [63]. In addition, the addition of trace element Sc in the Al alloy forms Al_3_(Sc, Zr) particles, hinders the slip of grain boundary dislocation and inhibits the recrystallization process, which reduces the quenching sensitivity [64].

The trace impurity elements Fe and Si in Al alloys also have a significant influence on the quenching sensitivity of Al alloy. When the content of Fe and Si increases, a high melting point phase is formed in the alloy. A large amount of Cu and Mg elements are consumed, which increases the Zn/Mg ratio in the alloy. As mentioned above, the increase of the Zn/Mg ratio inhibits the precipitation of heterogeneous nuclei, which will lead to a decrease in quenching sensitivity [65]. Similarly, the increase of trace elements Mn, Zr, Fe, and Si content will reduce the hardenability of Al alloy. In addition to the content of trace elements in an Al alloy affecting hardenability, the size of the sum of the atomic radius difference of various elements also has a certain effect on hardenability. It is found that, when the sum of the atomic radius difference of Al alloys is small, the influence of the increase of trace elements on the hardenability of Al alloys will become smaller. For example, when the content of trace elements such as Zr is low, the influence of alloy atomic radius difference on the hardenability of Al alloys is reduced [66].

## 5. Inclusions and Prospects

In general, the most widely used method for determining the hardenability of high-performance Al alloys is still mainly based on the end quenching. However, the traditional end quenching-hardness test method has some defects and is not standardized; the determination technology needs to be improved greatly, and most of them refer to the determination of steel hardenability to determine the hardenability of Al alloys. Currently, the improvement work carried out by professionals and researchers is primarily based on the material’s hardenability level, strength, volume, and other specific variables for unilateral improvement, or to improve the end quenching process. Moreover, other hardenability determination methods do not specify the process or are difficult to popularize. The research gap between these method defections on traditional end quenching and the unilateral improvement in the end quenching process was located in the accuracy and convenience of hardenability evaluation. In our opinion, the remedial steps for closing the gaps should be given both the evaluation system of hardenability by digital simulation and the digital management of hardenability test equipment for fine operation.

It can be seen that new methods and systems for determining the hardenability of high-performance Al alloys need to be further explored and discovered. Some researchers have initially defined the exploration path. A relatively complete hardenability evaluation system will be established in the near future. For example, Al-Li alloy hardenability data processing and characterization methods already have comprehensive evaluation methods, such as the most common method-hardness curve of ageing state, hardening sensitivity of solid solution state conductivity curve, mechanical properties related to tensile strength curve, temperature and time-related TTT and TTP curve, reflecting the change of microstructure and properties of CCT curve, and theoretical methods of calculation. Although these methods have initially constructed a system of characterization methods, the accuracy of these methods is still insufficient. It is necessary to further improve the existing characterization methods or invent some more accurate characterization methods to get more accurate experimental results. In addition, the factors affecting the hardenability of Al alloys includes the combination mode and the added content of main alloying elements and trace elements, which interact with each other to the adjusting of the hardenability. In the future, a variety of testing methods and characterization methods will be applied to Al alloys whose hardenability is controlled by the addition of microalloying elements, which will provide important technical support for the performance improvement of large structural components in the aerospace field.

## Figures and Tables

**Figure 1 materials-16-04736-f001:**
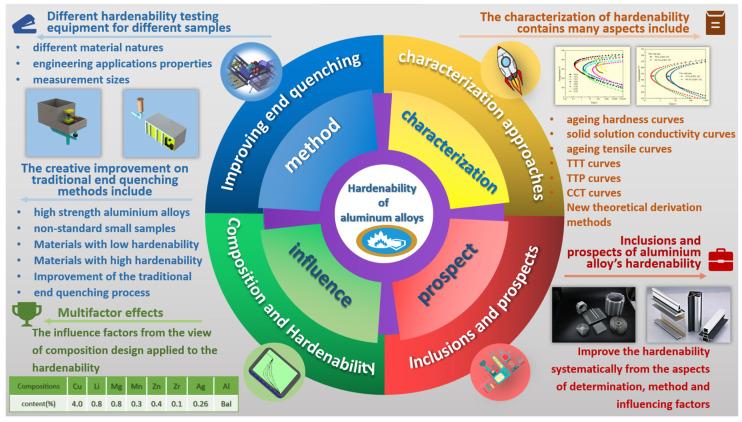
Schematic diagram of test and characterization of hardenability of aluminum alloys [18,19,20].

**Figure 2 materials-16-04736-f002:**
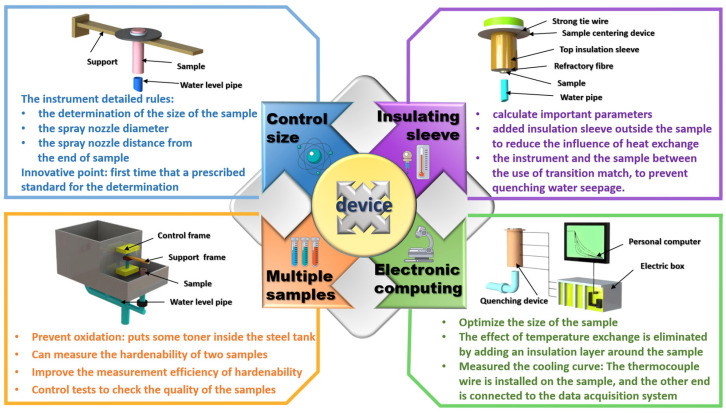
The new modified and developed applications of the end quenching method applied to the Al alloys [15,16,35,36].

**Figure 3 materials-16-04736-f003:**
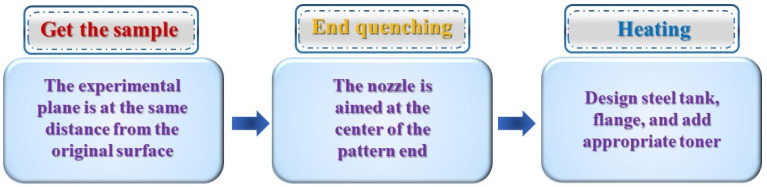
The process flow diagram of traditional and modified end quenching process [11].

**Figure 4 materials-16-04736-f004:**
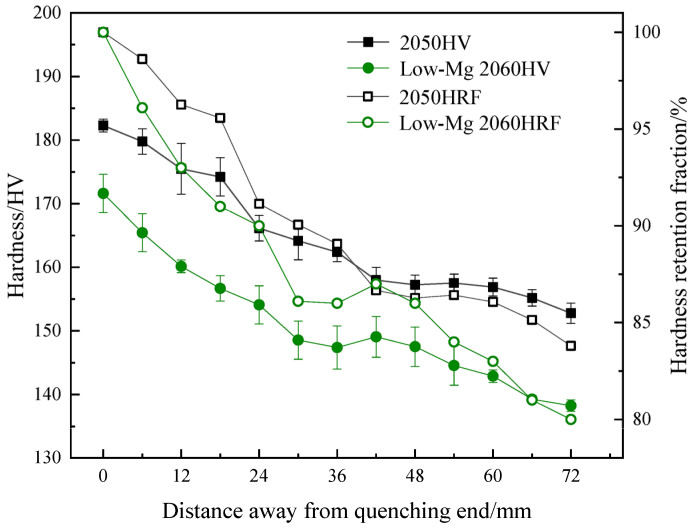
Relationship between hardness (HV), hardness retention fraction (HRF) and distance from the quenched end of high-performance aluminum alloys including the relatively high Mg content 2050 and low Mg content 2060 alloy [13,43].

**Figure 5 materials-16-04736-f005:**
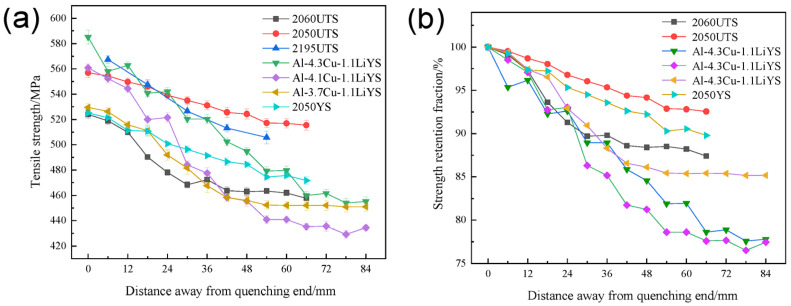
Relationship between (**a**) tensile strength and (**b**) retention fraction and distance from the quenched end of Al-Cu-Li Alloys [19,29,44].

**Figure 6 materials-16-04736-f006:**
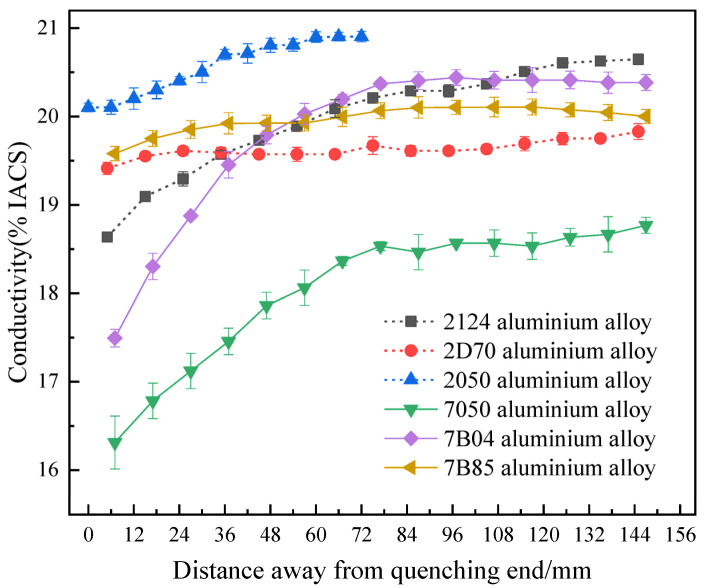
Relationship between conductivity and distance from the quenched end of different types of aluminum alloy [11,45].

**Figure 7 materials-16-04736-f007:**
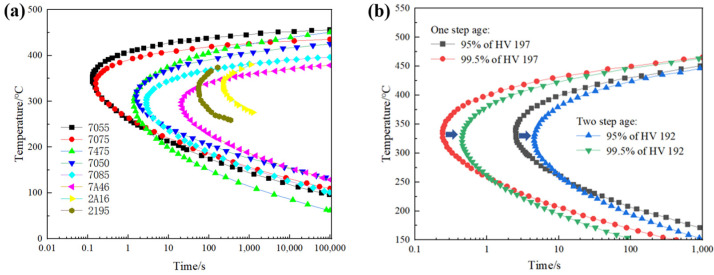
TTP fitting curves of aluminum alloys: (**a**) different chemical composition; (**b**) different multiple stage age [14,20,46].

**Figure 8 materials-16-04736-f008:**
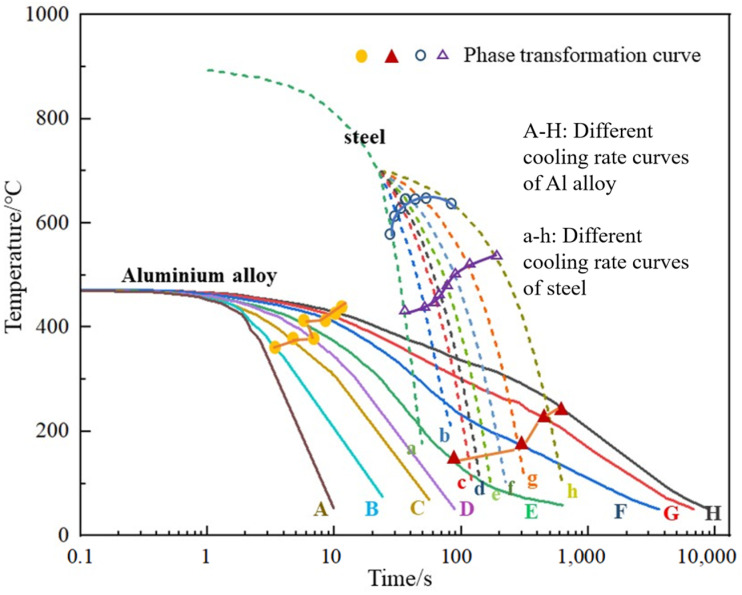
CCT diagram of continuous cooling transition in different chemical compositions of Aluminum Alloy [17,47].

**Figure 9 materials-16-04736-f009:**
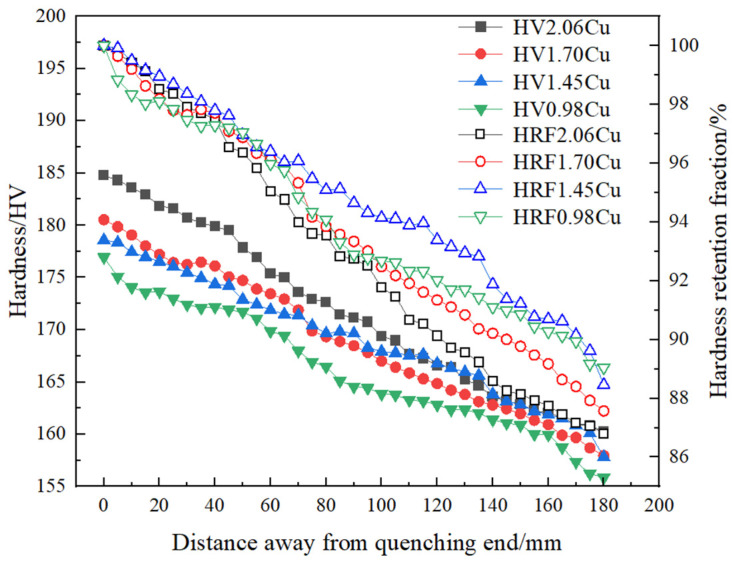
Hardenability and hardness retention curves of the aluminum alloy with different Cu contents [59,60,61].

**Table 1 materials-16-04736-t001:** The technical parameters details of the traditional end-quenching method for the hardenability testing of alloy [9,10,13,14,21,22,23,24,25,26,28,29,30,34].

Time	Quenching Method	Method Feature	Proposed Author
The first decades of the last century	The top quenching test or the Jominy end quenching test	Easy handling and repeatability	Jominy and Boege [21]
In recent years	Quantitative regulation of water column parameters	More precise	JoongKi, H. [24]
In recent years	The mechanical tolerance defects control	More precise	Dong, M.X. [34]
In recent years	the computer simulatation	more convenient and quick	Akatsuka, H. and Lin, J.H. [25,26]

**Table 2 materials-16-04736-t002:** Some end quenching methods for the aluminum alloy with different properties and different aluminum alloy samples [16,35,36].

Al Alloy Type	Pattern Size	Holding Method	Holding Time Drawing	How to Draw a Curve
High-strength Al alloy	Ø50 mm × 150 mm	Add insulation around the pattern bar	Control end quenching time	No data
Medium and low hardenability Al alloy	Ø80 mm, The height is half the thickness	No data	Need to be control	Strength curves
High hardenability Al alloy	Ø80 mm	Add a stainless steel insulation block	Need to be control	Hardness curve
Non-standard Al alloy sample	non-standard	Add insulation sleeve	Heat preservation for 12 min	End quenching curve

**Table 3 materials-16-04736-t003:** The current problems, solution approaches and proposed authors of the new quenching devices designed for the hardenability testing of aluminum alloy [39,40,41,42].

The Current Problems	Solution Approach	Improvement Targets	Proposed Author
Only one sample quenched at one time	A fixed frame installing a U-shaped tube with a water inlet at the bottom	Two or more samples quenched measurement at one time	Sun et al. [39]
Only one type of sample quenched at one time	An instrument using a combination way of metal sheet and end quenching	Two or more samples with different components quenched measurement at one time	Li et al. [40]
The quenching temperature control	the water spraying unit and drainage unit installed on the test cylinder	a device for accurate feedback and adjustment of quenching temperature	[41]
Specific types of Al alloy with relatively large predicted hardenability	The critical diameter method and the borehole plug method	Address the issue of erroneous hardenability values of alloys with high hardenability	[42]

**Table 4 materials-16-04736-t004:** Innovative test methods for the testing of the hardenability [11,15,16,35,36,37,38,39,40,41,42].

Problems When Testing the Hardenability	Improvement of Instruments
The data of the two test surfaces are different; The hardness distribution gradient is not reasonable	Design heating steel tank, non-quenched end design flange, and add toner
The traditional device has low mechanization and low efficiency	The workpiece is heated on the support plate, the hose water is cooled, and the data acquisition device is added
Few samples can be quenched at the same time, but two or more samples are needed	The U-shaped pipe is installed at the bottom of the pattern, and the water inlet is arranged at the bottom of the pipe, using the laminate end quenching method
Determine whether the quenching temperature meets the requirements	Water spraying and drainage units are installed on the test barrel

**Table 5 materials-16-04736-t005:** The current methods for the hardenability characterization of Aluminum alloys [13,14,15,16,17,18,19,20,21,22,23,24,25,26,27,28,29,30,31,32,33,34,35,36,37,38,39,40,41,42,43,44,45,46,47,48,49,50].

The Method	Application Alloys	Quenching Features	References
Hardness curve	Al-Zn-Mg-Cu and other quenched Al alloys	Hardness to Hardenability	[15]
Hardness curve	Aluminum alloy	Hardenability changes of Vickers hardness	[13,43]
Tensile strength curve	high-strength Al-Li alloys	Tensile strength curve to hardenability	[19,44,45]
Conductivity curve	the high-performance Al-Cu-Mg and Al-Zn-Mg-Cu alloys	The conductivity change to quenching sensitivity	[11]
TTT curve	Various types of alloys	Conductivity and hardness values to quenching sensitivity	[23]
TTP curve	Aluminum alloy	“C” curve to quenching sensitivity	[14,46]
CCT curve	Various types of alloys	DSC test, dynamic resistance test, XRD test and hardness test to hardenability evolution	[17,47]
The theoretical calculation method	Various types of alloys	A complementary means to accurately characterize the hardenability	[48,49,50]

**Table 6 materials-16-04736-t006:** The grain characteristic of statistical results of different samples with different Zn contents (in wt.%) [56].

Grain Structure Features	6%	7%	8%	9%
Recrystallization fraction (%)	18.3	23.3	28.6	39.3
HAGB (mm/mm^2^)	140.2	157.1	172.3 ± 16.3	185.5
LAGB (mm/mm^2^)	514.5	427.9	311.7 ± 19.8	155.5
HAGB + LAGB (mm/mm^2^)	654.7	585.0	484.0 ± 18.9	I341.0

## Data Availability

No new data were created or analyzed in this study. Data sharing is not applicable to this article.

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
