# Peer review of "Recent Progress in Testing and Characterization of Hardenability of Aluminum Alloys: A Review"

_materials, 2023, doi:10.3390/ma16134736_

Round 1

Reviewer 1 Report

-Figure should be bigger for clear interpretation.

-You can add table for each sections, which will show the authors, their process and the result.

-Describe the research gap you observed in AA processing and state the remedial steps.

Author Response

Dear Reviewer :

Thank you very much for giving us a chance for revision. We are grateful for these valuable comments from referees. We have fully comprehended these comments and have made many improvements which we hope meet with approval.

According to the referees’ comments, the main modifications have been made point-by-point as a file “response to referees”. Our responses are given in blue font in the “response to referees” and changes/additions to the manuscript are given in highlight text in the revised version. The responses and corresponding changes are shown as follow:

  1. Figure should be bigger for clear interpretation.

Response:

Thank you for the comment. In the revised version, Figure 1, Figure 2, Figure 4 and Figure 6 were magnification for the clear interpretation. At the same time, the text in the picture is also appropriately enlarged for clear display.

  1. You can add table for each sections, which will show the authors, their process and the result.

Response:

Thank you for your suggestions. We agree with referee’s opinion, Table 1, Table 3 and Table 5 were added to show the technical parameters details of the traditional end-quenching method for the hardenability testing, the current problems, solution approaches and proposed authors of the new quenching devices designed for the hardenability testing and the current methods for the hardenability characterization of Al alloys in the revised version.

Time

Quenching method

Method feature

Proposed author

The first decades of last century

The top quenching test or the Jominy end quenching test

Easy handling and repeatability

Jominy and Boege[21]

In recent years

Quantitative regulation of water column parameters

More precise

H. JoongKi [24]

In recent years

The mechanical tolerance defects control

More precise

M.X. Dong [34]

In recent years

the computer simulatation

more convenient and quick

H. Akatsuka and  J.H. Lin [25, 26]

Table 1. The technical parameters details of the traditional end-quenching method for the hardenability testing of alloy [9, 10, 13, 14, 21-26, 28-30, 34]

Table 3. The current problems, solution approaches and proposed authors of the new quenching devices designed for the hardenability testing of Aluminium alloy [39-42]

The current problems

Solution approach

Improvement targets

Proposed author

Only one sample quenched at one time

A fixed frame installing a U-shaped tube with a water inlet at the bottom

Two or more samples quenched measurement at one time

Sun et al.[39]

Only one type sample quenched at one time

An instrument using a combination way of metal sheet and end quenching

Two or more samples with different components quenched measurement  at one time

Li et al.[40]

The quenching temperature control

the water spraying unit and drainage unit  installed on the test cylinder

a device to accurate feedback and adjustment of quenching temperature

[41]

Specific types of Al alloy with relatively large predicted hardenability

The critical diameter method and the borehole plug method

Address the issue of erroneous hardenability values of alloys with high hardenability

[42]

Table 5. The current methods for the hardenability characterization of Aluminium alloys [13-50]

The method

Application alloys

Quenching features

References

Hardness curve

Al-Zn-Mg-Cu and other quenched Al alloys

Hardness to Hardenability

[15]

Hardness curve

Aluminium alloy

Hardenability changes of Vickers hardness

[13, 43]

Tensile strength curve

high-strength Al-Li alloys

Tensile strength curve to hardenability

[19, 44, 45]

Conductivity curve

the high-performance Al-Cu-Mg and Al-Zn-Mg-Cu alloys

The conductivity change to quenching sensitivity

[11]

TTT curve

Various types of alloys

Conductivity and hardness values to quenching sensitivity

[23]

TTP curve

Aluminium alloy

“C” curve to quenching sensitivity

[14, 46]

CCT curve

Various types of alloys

DSC test, dynamic re-sistance test, XRD test and hardness test to hardenability evolution

[17, 47]

The theoretical calculation method

Various types of alloys

A complementary means to accurately characterize the hardenability

[48, 49, 50]

  1. Describe the research gap you observed in AA processing and state the remedial steps

Response:

Thank you very much for pointing this out. In the updated version, the corresponding describe of the research gap and remedial steps were added in the Inclusions and Prospects section. The relative sentences were as below.

“Currently, the improvement work done by professionals and researchers is primarily based on the material's hardenability level, strength, volume, and other specific variables for unilateral improvement, or to improve the end quenching process. Moreover, other hardenability determination methods do not specify the process or are difficult to be popularized. The research gap between these method defections on traditional end quenching and the unilateral improvement in the end quenching process was located in the accuracy and convenience of hardenability evaluation. In our opinion, the remedial steps for closing the gaps should be given the evaluation System of hardenability by digital simulation and the Digital management of hardenability test equipment for fine operation.

Many other minor revisions have also been made in the revised manuscript to improve the flow and readability of the manuscript. We hope that these changes address the reviewers’ comments and feel that the resubmitted manuscript meets the criteria of significance in Materials.

Thank the referees for their help again. If you have further questions and comments concerning this manuscript, please feel free to contact me at any time.

Best regards,

Dan-yang Liu

Reviewer 2 Report

Manuscript "Recent Progress in Testing and Characterization of Hardenability of Aluminum Alloys: A Review" is devoted to the actual topic. The authors analyzed the existing methods and characterization approaches of hardenability for aluminum alloys. Review also drew attention to the current direction in the research of light materials in different areas. The work is of interest to a wide range of readers. The figures are clear, the work is clearly structured. 

I recommend accept the work in present form.

Author Response

Dear Reviewer:

Thank you very much for giving us a chance for revision. We are grateful for these valuable comments from referees. We have fully comprehended these comments and have made many improvements which we hope meet with approval.

According to the referees’ comments, the main modifications have been made point-by-point as a file “response to referees”. Our responses are given in blue font in the “response to referees” and changes/additions to the manuscript are given in highlight text in the revised version. The responses and corresponding changes are shown as follow:

  1. Line 34 rewrite "a nanometer-sized size"

Response:

Thank you for the comment. The sentence has been modified in line 32-34 of the updated version.

“Many heat-treatable alloy materials are heated above their solvus line to form the supersaturated solid solution, which is followed by the formation of a nanometer-sized strengthening precipitate [3].

  1. First comes a link to sources 2-5, and then a link to the first. It is necessary to number the references to the literature as they appear in the text. Also, the first source is quite often indicated at the end of the manuscript (subparagraph 4.1), so I recommend replacing the link to another article first, so that at the end of the article the first number does not stand out among other links.

Response:

Thank you for this suggestion. The links have been updated in the revised version.

  1. Line 40: "on alloy engineering application properties" can be removed or replaced with physico-chemical properties?

Line 44: “as testing, testing process.” how does one differ from the other?

Response:

Thank you for the comment. The sentence has been modified in line 38-41, line 43-44 of the updated version.

“It will be related to the evolution of solid phase transformation in unstable supersaturated solid solution alloys at a specific temperature, and it will also have a direct impact on physico-chemical properties, especially hardenability.

“The study on the hardenability of alloy materials is based on factors such as testing process and characterization improvement, influencing factors, and more.

  1. Line 86: Something happened to the reference [1, 6]. Also, 2-3 more sources should be added to it, since you speak of standards and norms in the plural.

Lines 86-90: supporting references to literary sources are needed.

- Add a link to Figure 1 before the actual figure.

- Line 109: link 7

Response:

Many thanks for your important comment. The more sources (3, 8, 9, 10, 11 and 12, 13, 14) were added in the mentioned sentence in the updated version. A link was added before the actual figure 1. The link was added in line 109.

“The distribution characteristics of the hardened depth layer and the evolution of hardness distribution of the test specimen can be used to express the hardenability characteristics of the tested material under the basic test conditions stipulated according to national standards or other scientific norms [3, 8, 9, 10, 11].

“However, nonferrous alloy metals, especially high-performance precipitation-strengthening Al alloys, have not obtained the actual and complete hardenability curve in the actual process of hardenability testing and characterization [12, 13, 14].”

  1. The introduction should end with the suggestion that the above needs and problems have been considered and analyzed in this paper.

Response:

Thank you for the comment. In the updated version, the last paragraph of the introduction section was revised to end with the suggestion that the above needs and problems.

For these influence factors affected quenching behaviors, some works on improvement of the test and characterization of the hardenability of Aluminum alloys were needed to develop. The special hardenability testing equipment was creative to suit the different types of alloys and quenching process, the new theoretical derivation methods were developed to characterize the overall quenching change feature, and the hardenability of determination method was systematically improved from the composition design of the quenching-resistance alloy by alloying and microalloying element control (shown in Fig. 1). Hence, it’s critical and meaningful to study the determination and characterization of the hardenability of high-performance Al alloy plates, as well as the influences of added element type and content on the hardenability property of Al alloy.

  1. Line 118 “performance uniformity of large-size metal structural components” please confirm with literature, maybe it's not so bad?

Line 150 “are not established [18]” It's been a long time, the search really doesn't show anything? It's just that it's not quite clear why this happened, and the reference to more than 10 years of research is mistakenly misleading the reader. I would understand it in such a way that this problem is not interesting now, but I'm not right?

Line 156: "quenched data" is very technically written.

Response:

Thank you for the comment. We have modified the sentences in lines 118, 150 and 156.

“Since the first few decades of the last century, the engineering field under the wave of the Industrial Revolution has continuously improved the requirements for the service performance of large-size metal structural components, which makes the exploration of scientific test methods for the hardenability of the metal materials plate become an inevitable trend.”

“Numerous technical parameters assessment system, including nozzle diameter, spray height, and sample size, which are suitable for an Al alloy hardenability test device, are not systematicly and completely established to the field of hardenability evaluation of high-performance aluminum alloys [13,14, 28, 29].”

“Before quenching, the technological process and quenching fixtures differ from actual production, making it impossible for typical industrial quenching to accurately represent the quenching properties by the data obtained from the quenching cooling process [30]”

  1. Expand on the sentence: “There are insufficient appropriate data conversion and characterization procedures used after experimental quenching when processing the data.”

Response:

Thank you for the comment. We have modified the sentence.

“There are insufficient appropriate data conversion and characterisation procedures used after quenching when processing the experimental data obtained from the quenching test of big sized structural alloy. The data conversion was usually used to the relation between the quenching sensitivity and quenching hardenability by the mechanical properties change as a function of the distance from the quenching end, while characterisation procedures was employed to determine the microstructural features evolution of the alloy materials sustained different quenching cooling conditions.”

  1. “Not the hardenability measuring of Aluminum alloy should be solved, but the end quenching standard should be refined” is this your suggestion to the scientific community? The link is obviously missing.

Response:

Thank you for the comment. We have modified the sentence.

“Some scholars proposed the primality point was not to solve the hardenability measuring but refine the end quenching standard to suit new type of alloy materials [31, 32, 33].”

  1. - The paragraph on lines 195-200 needs to be either significantly shortened or deleted.

Response:

Thank you for the comment. We have modified the sentence.

“At present, a wide demand for new high-performance Al alloy materials in industrial production gradually appears, which requires accurate hardenability data by modifying or improvement on traditional end quenching tha the traditional method can’t obtain.”

  1. - Fig. 2 move to subparagraph 2.2.

- Add a link before table 1.

Response:

Thank you for the comment. Figure 2 have been moved to subparagraph 2.2. And a link was added before table 1.

  1. - Check the link text in Fig. 4-9, they must be specified before the figures appear. Name of Fig. 4 to be supplemented, since the hardenability is also presented on it. On Fig. 5 increase the font at the axes by 1.5-2 times. Title for Fig. 6 is not aligned correctly.

Response:

Thank you for the comment. We have modified these Figures. The link text was checked in Fig. 4-9. Name of Fig. 4 was revised. The font at the axes of Fig. 5 was increased to 1.5-2 times. And the title of the Fig. 6 was corrected in the updated version.

Figure 4. Relationship between hardness (HV), hardness retention fraction(HRF) and distance from the quenched end of high-performance Aluminum Alloys including the relatively high Mg content 2050 and low Mg content 2060 alloy [13, 43]

Fig. 6 Relationship between conductivity and distance from the quenched end of different type of Aluminum Alloy [11, 45]”

  1. - Check the list of references carefully. References 1, 9, 10, 19. Reference 35 contains the only DOI. Although it is more, correct to indicate DOI to all sources that have it.

Response:

Thank you for the comment. We have checked and corrected the all references in the updated version.

Many other minor revisions have also been made in the revised manuscript to improve the flow and readability of the manuscript. We hope that these changes address the reviewers’ comments and feel that the resubmitted manuscript meets the criteria of significance in Materials.

Thank the referees for their help again. If you have further questions and comments concerning this manuscript, please feel free to contact me at any time.

Best regards,

Dan-yang Liu

Reviewer 3 Report

The manuscript "Recent Progress in Testing and Characterization of Hardenability of Aluminum Alloys: A Review" has good potential in scientific terms and high practical significance. The analysis carried out is worthy of publication.

However, the following remarks to the manuscript:

- Line 34 rewrite "a nanometer-sized size".

- First comes a link to sources 2-5, and then a link to the first. It is necessary to number the references to the literature as they appear in the text. Also, the first source is quite often indicated at the end of the manuscript (subparagraph 4.1), so I recommend replacing the link to another article first, so that at the end of the article the first number does not stand out among other links.

- Line 40: "on alloy engineering application properties" can be removed or replaced with physico-chemical properties?

- Line 44: “as testing, testing process.” how does one differ from the other?

- Line 86: Something happened to the reference [1, 6]. Also, 2-3 more sources should be added to it, since you speak of standards and norms in the plural.

- Lines 86-90: supporting references to literary sources are needed.

- Add a link to Figure 1 before the actual figure.

- Line 109: link 7.

- The introduction should end with the suggestion that the above needs and problems have been considered and analyzed in this paper…

- Line 118 “performance uniformity of large-size metal structural components” please confirm with literature, maybe it's not so bad?

- Line 150 “are not established [18]” It's been a long time, the search really doesn't show anything? It's just that it's not quite clear why this happened, and the reference to more than 10 years of research is mistakenly misleading the reader. I would understand it in such a way that this problem is not interesting now, but I'm not right?

- Line 156: "quenched data" is very technically written.

- Expand on the sentence: “There are insufficient appropriate data conversion and characterization procedures used after experimental quenching when processing the data.”

- “Not the hardenability measuring of Aluminum alloy should be solved, but the end quenching standard should be refined” is this your suggestion to the scientific community? The link is obviously missing.

- The paragraph on lines 195-200 needs to be either significantly shortened or deleted.

- Fig. 2 move to subparagraph 2.2.

- Add a link before table 1.

- Check the link text in Fig. 4-9, they must be specified before the figures appear. Name of Fig. 4 to be supplemented, since the hardenability is also presented on it. On Fig. 5 increase the font at the axes by 1.5-2 times. Title for Fig. 6 is not aligned correctly.

- Check the list of references carefully. References 1, 9, 10, 19. Reference 35 contains the only DOI. Although it is more, correct to indicate DOI to all sources that have it.

Author Response

Dear Mrs Tamara Stojković:

Thank you very much for giving us a chance for revision. We are grateful for these valuable comments from referees. We have fully comprehended these comments and have made many improvements which we hope meet with approval.

According to the referees’ comments, the main modifications have been made point-by-point as a file “response to referees”. Our responses are given in blue font in the “response to referees” and changes/additions to the manuscript are given in highlight text in the revised version. The responses and corresponding changes are shown as follow:

Reviewer 1

  1. Figure should be bigger for clear interpretation.

Response:

Thank you for the comment. In the revised version, Figure 1, Figure 2, Figure 4 and Figure 6 were magnification for the clear interpretation. At the same time, the text in the picture is also appropriately enlarged for clear display.

  1. You can add table for each sections, which will show the authors, their process and the result.

Response:

Thank you for your suggestions. We agree with referee’s opinion, Table 1, Table 3 and Table 5 were added to show the technical parameters details of the traditional end-quenching method for the hardenability testing, the current problems, solution approaches and proposed authors of the new quenching devices designed for the hardenability testing and the current methods for the hardenability characterization of Al alloys in the revised version.

Time

Quenching method

Method feature

Proposed author

The first decades of last century

The top quenching test or the Jominy end quenching test

Easy handling and repeatability

Jominy and Boege[21]

In recent years

Quantitative regulation of water column parameters

More precise

H. JoongKi [24]

In recent years

The mechanical tolerance defects control

More precise

M.X. Dong [34]

In recent years

the computer simulatation

more convenient and quick

H. Akatsuka and  J.H. Lin [25, 26]

Table 1. The technical parameters details of the traditional end-quenching method for the hardenability testing of alloy [9, 10, 13, 14, 21-26, 28-30, 34]

Table 3. The current problems, solution approaches and proposed authors of the new quenching devices designed for the hardenability testing of Aluminium alloy [39-42]

The current problems

Solution approach

Improvement targets

Proposed author

Only one sample quenched at one time

A fixed frame installing a U-shaped tube with a water inlet at the bottom

Two or more samples quenched measurement at one time

Sun et al.[39]

Only one type sample quenched at one time

An instrument using a combination way of metal sheet and end quenching

Two or more samples with different components quenched measurement  at one time

Li et al.[40]

The quenching temperature control

the water spraying unit and drainage unit  installed on the test cylinder

a device to accurate feedback and adjustment of quenching temperature

[41]

Specific types of Al alloy with relatively large predicted hardenability

The critical diameter method and the borehole plug method

Address the issue of erroneous hardenability values of alloys with high hardenability

[42]

Table 5. The current methods for the hardenability characterization of Aluminium alloys [13-50]

The method

Application alloys

Quenching features

References

Hardness curve

Al-Zn-Mg-Cu and other quenched Al alloys

Hardness to Hardenability

[15]

Hardness curve

Aluminium alloy

Hardenability changes of Vickers hardness

[13, 43]

Tensile strength curve

high-strength Al-Li alloys

Tensile strength curve to hardenability

[19, 44, 45]

Conductivity curve

the high-performance Al-Cu-Mg and Al-Zn-Mg-Cu alloys

The conductivity change to quenching sensitivity

[11]

TTT curve

Various types of alloys

Conductivity and hardness values to quenching sensitivity

[23]

TTP curve

Aluminium alloy

“C” curve to quenching sensitivity

[14, 46]

CCT curve

Various types of alloys

DSC test, dynamic re-sistance test, XRD test and hardness test to hardenability evolution

[17, 47]

The theoretical calculation method

Various types of alloys

A complementary means to accurately characterize the hardenability

[48, 49, 50]

  1. Describe the research gap you observed in AA processing and state the remedial steps

Response:

Thank you very much for pointing this out. In the updated version, the corresponding describe of the research gap and remedial steps were added in the Inclusions and Prospects section. The relative sentences were as below.

“Currently, the improvement work done by professionals and researchers is primarily based on the material's hardenability level, strength, volume, and other specific variables for unilateral improvement, or to improve the end quenching process. Moreover, other hardenability determination methods do not specify the process or are difficult to be popularized. The research gap between these method defections on traditional end quenching and the unilateral improvement in the end quenching process was located in the accuracy and convenience of hardenability evaluation. In our opinion, the remedial steps for closing the gaps should be given the evaluation System of hardenability by digital simulation and the Digital management of hardenability test equipment for fine operation.

Reviewer 2

  1. Line 34 rewrite "a nanometer-sized size"

Response:

Thank you for the comment. The sentence has been modified in line 32-34 of the updated version.

“Many heat-treatable alloy materials are heated above their solvus line to form the supersaturated solid solution, which is followed by the formation of a nanometer-sized strengthening precipitate [3].

  1. First comes a link to sources 2-5, and then a link to the first. It is necessary to number the references to the literature as they appear in the text. Also, the first source is quite often indicated at the end of the manuscript (subparagraph 4.1), so I recommend replacing the link to another article first, so that at the end of the article the first number does not stand out among other links.

Response:

Thank you for this suggestion. The links have been updated in the revised version.

  1. Line 40: "on alloy engineering application properties" can be removed or replaced with physico-chemical properties?

Line 44: “as testing, testing process.” how does one differ from the other?

Response:

Thank you for the comment. The sentence has been modified in line 38-41, line 43-44 of the updated version.

“It will be related to the evolution of solid phase transformation in unstable supersaturated solid solution alloys at a specific temperature, and it will also have a direct impact on physico-chemical properties, especially hardenability.

“The study on the hardenability of alloy materials is based on factors such as testing process and characterization improvement, influencing factors, and more.

  1. Line 86: Something happened to the reference [1, 6]. Also, 2-3 more sources should be added to it, since you speak of standards and norms in the plural.

Lines 86-90: supporting references to literary sources are needed.

- Add a link to Figure 1 before the actual figure.

- Line 109: link 7

Response:

Many thanks for your important comment. The more sources (3, 8, 9, 10, 11 and 12, 13, 14) were added in the mentioned sentence in the updated version. A link was added before the actual figure 1. The link was added in line 109.

“The distribution characteristics of the hardened depth layer and the evolution of hardness distribution of the test specimen can be used to express the hardenability characteristics of the tested material under the basic test conditions stipulated according to national standards or other scientific norms [3, 8, 9, 10, 11].

“However, nonferrous alloy metals, especially high-performance precipitation-strengthening Al alloys, have not obtained the actual and complete hardenability curve in the actual process of hardenability testing and characterization [12, 13, 14].”

  1. The introduction should end with the suggestion that the above needs and problems have been considered and analyzed in this paper.

Response:

Thank you for the comment. In the updated version, the last paragraph of the introduction section was revised to end with the suggestion that the above needs and problems.

For these influence factors affected quenching behaviors, some works on improvement of the test and characterization of the hardenability of Aluminum alloys were needed to develop. The special hardenability testing equipment was creative to suit the different types of alloys and quenching process, the new theoretical derivation methods were developed to characterize the overall quenching change feature, and the hardenability of determination method was systematically improved from the composition design of the quenching-resistance alloy by alloying and microalloying element control (shown in Fig. 1). Hence, it’s critical and meaningful to study the determination and characterization of the hardenability of high-performance Al alloy plates, as well as the influences of added element type and content on the hardenability property of Al alloy.

  1. Line 118 “performance uniformity of large-size metal structural components” please confirm with literature, maybe it's not so bad?

Line 150 “are not established [18]” It's been a long time, the search really doesn't show anything? It's just that it's not quite clear why this happened, and the reference to more than 10 years of research is mistakenly misleading the reader. I would understand it in such a way that this problem is not interesting now, but I'm not right?

Line 156: "quenched data" is very technically written.

Response:

Thank you for the comment. We have modified the sentences in lines 118, 150 and 156.

“Since the first few decades of the last century, the engineering field under the wave of the Industrial Revolution has continuously improved the requirements for the service performance of large-size metal structural components, which makes the exploration of scientific test methods for the hardenability of the metal materials plate become an inevitable trend.”

“Numerous technical parameters assessment system, including nozzle diameter, spray height, and sample size, which are suitable for an Al alloy hardenability test device, are not systematicly and completely established to the field of hardenability evaluation of high-performance aluminum alloys [13,14, 28, 29].”

“Before quenching, the technological process and quenching fixtures differ from actual production, making it impossible for typical industrial quenching to accurately represent the quenching properties by the data obtained from the quenching cooling process [30]”

  1. Expand on the sentence: “There are insufficient appropriate data conversion and characterization procedures used after experimental quenching when processing the data.”

Response:

Thank you for the comment. We have modified the sentence.

“There are insufficient appropriate data conversion and characterisation procedures used after quenching when processing the experimental data obtained from the quenching test of big sized structural alloy. The data conversion was usually used to the relation between the quenching sensitivity and quenching hardenability by the mechanical properties change as a function of the distance from the quenching end, while characterisation procedures was employed to determine the microstructural features evolution of the alloy materials sustained different quenching cooling conditions.”

  1. “Not the hardenability measuring of Aluminum alloy should be solved, but the end quenching standard should be refined” is this your suggestion to the scientific community? The link is obviously missing.

Response:

Thank you for the comment. We have modified the sentence.

“Some scholars proposed the primality point was not to solve the hardenability measuring but refine the end quenching standard to suit new type of alloy materials [31, 32, 33].”

  1. - The paragraph on lines 195-200 needs to be either significantly shortened or deleted.

Response:

Thank you for the comment. We have modified the sentence.

“At present, a wide demand for new high-performance Al alloy materials in industrial production gradually appears, which requires accurate hardenability data by modifying or improvement on traditional end quenching tha the traditional method can’t obtain.”

  1. - Fig. 2 move to subparagraph 2.2.

- Add a link before table 1.

Response:

Thank you for the comment. Figure 2 have been moved to subparagraph 2.2. And a link was added before table 1.

  1. - Check the link text in Fig. 4-9, they must be specified before the figures appear. Name of Fig. 4 to be supplemented, since the hardenability is also presented on it. On Fig. 5 increase the font at the axes by 1.5-2 times. Title for Fig. 6 is not aligned correctly.

Response:

Thank you for the comment. We have modified these Figures. The link text was checked in Fig. 4-9. Name of Fig. 4 was revised. The font at the axes of Fig. 5 was increased to 1.5-2 times. And the title of the Fig. 6 was corrected in the updated version.

Figure 4. Relationship between hardness (HV), hardness retention fraction(HRF) and distance from the quenched end of high-performance Aluminum Alloys including the relatively high Mg content 2050 and low Mg content 2060 alloy [13, 43]

Fig. 6 Relationship between conductivity and distance from the quenched end of different type of Aluminum Alloy [11, 45]”

  1. - Check the list of references carefully. References 1, 9, 10, 19. Reference 35 contains the only DOI. Although it is more, correct to indicate DOI to all sources that have it.

Response:

Thank you for the comment. We have checked and corrected the all references in the updated version.

Many other minor revisions have also been made in the revised manuscript to improve the flow and readability of the manuscript. We hope that these changes address the reviewers’ comments and feel that the resubmitted manuscript meets the criteria of significance in Materials.

Thank the referees for their help again. If you have further questions and comments concerning this manuscript, please feel free to contact me at any time.

Best regards,

Dan-yang Liu

Round 2

Reviewer 1 Report

All the revisions are ok. Please draft the manuscript as per journal's format.

Author Response

Dear Reviewer:

  Thank you very much for giving us a chance for minor revision. We have fully comprehended your comments and have made some improvements to meet the journal's format requirement.

   Thank the referees for their help again. If you have further questions or comments concerning this manuscript, please feel free to contact me at any time.

  Best regards,

  Dan-yang Liu